# Game Birds Can Act as Intermediaries of Virulent Genotype VII Avian Orthoavulavirus-1 between Wild Birds and Domestic Poultry

**DOI:** 10.3390/v15020536

**Published:** 2023-02-14

**Authors:** Craig S. Ross, Paul Skinner, David Sutton, Jo Mayers, Alex Nunez, Sharon M. Brookes, Ashley C. Banyard, Ian H. Brown

**Affiliations:** Animal and Plant Health Agency (APHA), Woodham Lane, Addlestone KT15 3NB, UK

**Keywords:** Avian orthoavulavirus-1, AOAV-1, Newcastle disease, NDV, game birds, poultry

## Abstract

Newcastle Disease (ND), caused by virulent forms of Avian orthoavulavirus serotype-1 (AOAV-1) is an economically important avian disease worldwide. The past two incursions of ND into the United Kingdom occurred in game bird populations during 2005 and 2006. The nature of the game bird semi-feral rearing system, which can bring these birds into close contact with both wild birds and commercial or backyard poultry, has been hypothesized to act as a bridge between these two environments. As such, the risk that AOAV-1-infected game birds may pose to the UK poultry industry was investigated. Pheasants, partridges and chickens were experimentally infected with the virulent strain APMV-1/Chicken/Bulgaria/112/13, a genotype VII.2 virus associated with ND outbreaks in Eastern Europe. The study demonstrated that both chickens and pheasants are susceptible to infection with APMV-1/Chicken/Bulgaria/112/13, which results in high mortality and onward transmission. Partridges by contrast are susceptible to infection, but mortality was reduced, as was onward transmission. However, the data indicated that both pheasants and partridges may serve as intermediate hosts of AOAV-1 and may bridge the wild bird–domestic poultry interface enabling transmission into an economically damaging environment where morbidity and mortality may be high.

## 1. Introduction

Newcastle disease (ND) is an avian systemic infection caused by virulent strains of Avian orthoavulavirus (AOAV-1) (previously known as Avian paramyxovirus-1 (APMV-1) and Avian avulavirus-1 (AAvV-1) and [1]). The causative agent, termed Newcastle Disease Virus (NDV) where clinical definitions are met (OIE/WOAH, 2021), is an economically important disease of poultry worldwide. NDV is enzootic in many parts of the world and has a devastating impact on the global poultry industry due to its high flock mortality rates, which may result in trade restrictions during ongoing outbreak situations [2].

AOAV-1 virions contain single-stranded, non-segmented negative-strand RNA genomes of ca. 15,192 bp encoding for six structural proteins: nucleoprotein (NP), phosphoprotein (P), matrix protein (M), fusion protein (F), haemagglutinin-neuraminidase (HN) and the large polymerase protein (L), plus two accessory proteins V and W which result from the editing of the P gene [3]. AOAV-1 has been detected in 241 different bird species globally, within 27 orders [4], but is most commonly detected in domesticated Galliformes. Clinical signs of AOAV-1 infection, which sometimes lead to the development of severe clinical disease termed ND, are wide ranging and can vary between hosts. Even within an experimental setting, significant differences in infection outcome can occur in different poultry species [5,6]. The spectrum of disease within chickens falls into two broad phenotypes: (i) avirulent and lentogenic infection, which may cause asymptomatic infection or mild disease; and (ii) virulent infections whereby the outcome of disease can be moderate morbidity and low mortality (mesogenic) or severe morbidity and high mortality (velogenic). Both host and viral factors are thought to be involved in the outcome of infection, although the drivers for clinical disease outcomes remain poorly understood. However, although the factors that determine if an isolate is mesogenic or velogenic are undefined, it is well-established that the cleavage site (CS) of the F protein, and, in particular, the presence of multiple basic amino acids, is a critical factor in determining an isolate’s potential virulence [7]. The inactive F0 form of the Fusion protein must be cleaved into the active F1 and F2 forms in order for the AOAV-1 virion to successfully fuse with the cell membrane and enter the cytoplasm to release its genome and initiate the viral life cycle [8]. Avirulent and lentogenic viruses typically possess a CS containing fewer basic amino acids and a leucine at position 117. This form of motif is restricted to cleavage by trypsin-like proteases [9] found in the respiratory and digestive tissues, thus limiting viral replication to those areas. Virulent isolates that may cause either mesogenic or velogenic clinical signs, have a multi-basic amino acid CS with a phenylalanine at position 117 [7,10]. This motif is cleaved by Furin-like enzymes, which are found in most animal tissues [11] and allows for the establishment of a systemic infection, including the infection of the central nervous system, a factor that results in neurological sequelae. Due to the correlation of virulence with the F CS and significant disease [12], the detection of a virulent CS motif is one of the criteria that enables official designation as infection with NDV, according to the World Organization for Animal Health (WOAH) [13].

Although a virulent CS motif allows for systemic infection, it does not necessarily dictate the degree of clinical disease that may be seen. Therefore, to determine the pathogenicity of a virus experimentally, in vivo techniques must be relied upon. The intracerebral pathogenicity index (ICPI) is the WOAH-accepted method to conclusively determine the pathogenicity of an isolate. Those isolates having an ICPI score >0.7 and/or containing a virulent F cleavage site (CS) are classified as NDV [13] and should be reported to the WOAH.

The most recent outbreaks of NDV that occurred in the United Kingdom (UK) were detected in game birds in 2005 and in 2006. In 2005, a notable outbreak of NDV occurred at a property in Surrey farming common pheasants (*Phasianus colchicus*), in England. Five out of seven occupied pens (approximately 10,000 birds) developed clinical disease with mortality observed (<3%). The origin of the infection was later traced to chicks imported from a French breeder (Aldous et al., 2007). Two isolates recovered from the event were class II, genotype XIII.1.1 (previously lineage 5b, genotype XIIIa) strains, both possessing a virulent cleavage site with ICPI scores of 1.26 and 1.6, respectively [14]. Related genotype XIII.1.1 AOAV-1 viruses were also circulating in Finland, Denmark, and Sweden between 2002–2004, all with virulent CS and ICPI values > 1.3 [15]. The second outbreak occurred in 2006 in West Lothian, Scotland in grey partridges (*Perdix perdix*), and was caused by a genotype VI (lineage 4) virus, commonly associated with Pigeon paramyxovirus (PPMV-1). This isolate had an ICPI value of 1.01 [16]. The detection of virulent AOAV-1 in UK game birds raised concerns regarding their potential role as virus reservoirs.

Infections of pheasants with AOAV-1 have been reported from the 1940s onwards (for review see [17]). More recently, along with the two UK outbreaks, virulent genotype VII AOAV-1 infections in pheasants have been noted in Pakistan [18]. Serological analyses of wild and pen-reared pheasants have also confirmed the presence of antibodies against AOAV-1 in both Europe and the USA [19,20], demonstrating the natural avirulent infection of pheasants. Due to the semi-feral rearing processes within the sector, game birds may encounter wild birds, and indirectly contact intensively farmed poultry. There is limited information regarding the infections of partridges with AOAV-1, other than the UK outbreak, although velogenic ND has been noted in partridges at US customs [21]. Experimental infections of game birds with genotype VI.2 AOAV-1 (commonly associated with PPMV-1) has demonstrated differing shedding and transmission dynamics between pheasant and partridges [22]. Furthermore, the game bird industry involves the international movement of birds and their products, activities that have been previously linked to ND outbreaks [14].

To date, all AOAV-1 genotype VII viruses detected have been shown to contain a virulent F protein cleavage site, and those assessed so far have a virulent pathotype resulting in high mortality rates. These viruses are commonly found in the Far and Middle East, Africa and Europe [23,24] and outbreaks often result in large economic losses. Genotype VII outbreaks have recently been detected in Bulgaria, Turkey and Georgia [25] and in Belgium, the Netherlands and Luxemburg in 2018 [26] causing significant economic losses. These continual incursions demonstrate the threat of AOAV-1, even in countries with generally robust biosecurity measures.

To determine the potential role that game birds could play in transmission of virulent forms of AOAV-1, and any pathogenesis observed, the infection of pheasants, partridges and chickens was assessed. The birds were challenged with APMV-1/chicken/Bulgaria/112/13, a highly virulent genotype VII.2 virus (ICPI 1.93) as an example of AOAV-1 strains currently circulating in Europe (Fuller et al., 2017).

## 2. Materials and Methods

### 2.1. Viral Isolate

APMV-1/chicken/Bulgaria/112/13 (AV112-13), a genotype VII.2 (ICPI—1.93) was isolated from domestic fowl in Bulgaria in 2013 [25] and was propagated using 9–11 day old embryonated fowls’ eggs (EFEs) from specific-pathogen-free (SPF) flock.

### 2.2. Birds

The three bird species used in the study are as follows: 35 red-legged partridges (*Alectoris rufa*), 35 common ring-necked pheasants (*Phasianus colchicus*) and 29 SPF White Leghorn chickens (*Gallus gallus*) all 6–8 weeks of age. All birds were screened for anti-AOAV-1 antibodies using a hemagglutination inhibition (HI) test (Villegas, 2006). Housing and treatments were as described previously [22].

### 2.3. Experimental Design

The study was separated into two parts. The first element examined clinical presentation, infection dynamics and a small-scale transmission study of AV112-13 in a cohort of 15 partridges, 15 pheasants and 15 chickens at a dose of 1x106 EID50. Two in-contact birds per species were introduced and housed with their conspecifics at 1 day post infection (dpi) for examination of transmission. Prior to the commencement of the study, the birds were swabbed, and blood was sampled to confirm that there had been no prior exposure to AOAV-1 and were negative for infection. Buccal and cloacal swabs, and bird weights were taken daily. Two birds per pheasant and partridge species were killed humanely at 2, 4, 7, 9 and 11 dpi, whilst two chickens were killed humanely at 1, 2, 3, 4 and 5 dpi. This staggered euthanasia resulted in a decrease in birds throughout the study. Upon termination, brain, breast muscle, cecal tonsil, feather shafts, heart, intestine, kidney, liver, lung, spleen, trachea and turbinate tissues were collected and analyzed for the presence of viral antigen. The experiments were continued for a maximum of 21 days or were terminated earlier if no shedding was observed for two consecutive days in all directly infected birds. At the point of euthanasia, terminal heart bleeds were carried out for blood sampling, along with the recording of end-point weights and terminal swab collections. Terminal heart bleed however could not be performed on birds that had succumbed to infection.

The second element examined the transmission of AV112-13 between game bird conspecifics, and game birds and chickens, respectively. Briefly, six pheasants and partridges were directly inoculated with 1x106 EID50 of AV112-13 intranasally and intraocularly (Donor group 0—D0). The following day, six naïve conspecifics were introduced (Receiver group 1—R1) and periodic sampling was carried out to determine transmission to R1 birds. Upon detection of viral vRNA within swabs taken from the R1 group, the D0 birds were removed, and six further naïve conspecifics were introduced (R2). Transmission and swabbing from the buccal route were periodically undertaken to determine the presence of vRNA. If vRNA was detected in the R2 group, the R1 group was removed, and six naïve chickens introduced (R3).

### 2.4. Molecular Methods

Swabs were taken for molecular testing and processed as described previously [22]. Total nucleic acid was extracted from swab samples, which were then assessed using the molecular assays as described previously [27] with Ct values < 37 being considered AOAV-1 positive. A ten-fold serial dilution of titrated AV112-13 RNA was used to construct a standard curve using MxPro software (Stratagene), with Ct values converted to relative expressed units (REU) by correlation with EID50/mL of extracted viral standard on the curve [28]. In the case of the transmission study, the results are expressed as a 40-Ct value.

### 2.5. Serology

Hemagglutination Inhibition (HI) tests were carried out on serum samples using AV112-13 as the test antigen for HI as described by Villegas et al. (2006).

### 2.6. Histopathology and Immunohistochemistry (IHC)

The samples for histopathology were fixed in 10% neutral buffered formalin for a minimum of 5 days before being processed and embedded in paraffin. The sections for immunohistochemistry were prepared as described previously [29]. The antigen was retrieved using a heat treatment in pH9 buffer and visualized using ENVISION polymer and DAB (Dako™).

### 2.7. Statistical Analysis

The total nucleic acid shed was determined by area under the curve (AUC) analysis of the five birds intended to be kept until the end of the study. Statistical analysis was carried out using the Kruskall–Wallis test using GraphPad Prism v8 software (GraphPad Software, Inc., San Diego, CA, USA).

## 3. Results

### 3.1. Pathogenesis and Transmission of APMV-1/Chicken/Bulgaria/112/13 in Chickens, Pheasants and Partridges

To examine the pathogenesis of AV112-13 in the three avian species (*G. gallus*, *P. colchicus* and *A. rufa*), 15 birds from each species were directly infected with a high titer of AV112-13 (10^6^ EID_50_) intraocularly and intranasally. Due to the high ICPI score (1.93) and analysis of previous data [30] of highly virulent AOAV-1 in chickens, two chickens were removed at 1, 2, 3, 4 and 5 dpi for a post-mortem examination (PME) and an analysis of tissue distribution of the virus. A similar approach was taken for the game birds with two birds being removed at 2, 4, 7, 9 and 11 dpi for PME and tissue distribution analysis.

Following the direct inoculation of chickens, clinical signs occurred within 2 dpi, with diarrhea observed in five birds beginning at 2 dpi and affecting 56% of birds (n = 5/9) at 4 dpi and 80% (n = 4/5) at 5 dpi, alongside a generalized mucoid discharge from eyes and nostrils. Weight loss (Figure 1b) of 15–20% was observed (for those birds not selected for PME) with clinical disease including periocular oedema, huddling, drooped wings, depression and a reluctance to eat and drink. A loss of balance with tremors affected two birds (#485 and #494) from 4 dpi onwards with 100% mortality observed by 7 dpi (Figure 1a). Mean death time (MDT) of directly infected chickens (not selected for PME) with AV112-13 was 5.4 days.

The disease progression in AV112-13 infected pheasants was similar to that seen in chickens. Again, there was a rapid onset of clinical signs, with weight loss observed from 2 dpi, and a 15–20% drop in weight over the time-course (Figure 1c). Clinical observations in pheasants also included mild periocular oedema, huddling, closed eyes and drooped wings. Diarrhea was observed for three out of the thirteen birds (23%) by 4 dpi. Incoordination leading to loss of balance was observed in three birds at 4 dpi and in a further two birds by 5 dpi. Two of these latter birds also presented with more severe neurological signs, including torticollis. Due to the severe nature of the disease, all directly infected pheasants were killed humanely on welfare grounds by 7 dpi (Figure 1a), with an MDT of 5.6 days.

The clinical outcome of infection in partridges was markedly different to the other two bird species examined. No obvious clinical signs were observed until 5 dpi with all birds maintaining their body weight up to this point (Figure 1d). At 6 dpi, one bird (#102) was humanely killed following a rapid onset of clinical disease (huddling, ocular and oral discharge, depression and loss of balance/incoordination). Two birds (#101 and #104) exhibited a depressed state from 7 dpi, progressing to overt neurological disease including incoordination, ataxia, tremors and rapid weight loss resulting in their humane termination at 9 dpi. Of the birds not scheduled for post-mortem at 11 dpi, one had no signs of overt clinical disease with the only noticeable observation being its weight plateauing between 7 and 13 dpi. Another bird began to lose weight at 7 dpi and exhibited drooped wings, depression and a loss of balance from 9–11 dpi, however, by 16 dpi its weight had stabilized, clinical progression ceased, and its general condition continued to improve until the conclusion of the study. MDT could not be determined as two birds survived infection.

### 3.2. Viral Shedding Dynamics of AV112-13 Infection in Chickens, Pheasants and Partridges

To determine levels of viral nucleic acid shedding (as a surrogate for viral shedding), vRNA excretion was assessed by rRT-PCR. Analysis of swabs taken from chickens demonstrated the detection of vRNA from the buccal swab at 1 dpi (100% n = 15/15), although this could represent detection of residual viral nucleic acid from the inoculation site. From buccal swabs, the levels of vRNA were detected at the majority of timepoints, with levels increasing until 3 dpi before a decrease was observed until the termination of all birds at 7 dpi (Figure 2a). The analysis of cloacal vRNA showed detection at 2 dpi (62%; n = 8/13), with levels increasing, and all birds shedding vRNA by 3 dpi. vRNA detection from swabs continued until all birds had been humanely killed/succumbed to disease at 7 dpi (Figure 2a).

Analysis of pheasant buccal shedding was again, similar to chickens, with levels of detected vRNA increasing over time from 2 dpi (40%; n = 6/15) until all birds had been humanely killed/succumbed to infection by 7 dpi, with 100% of birds shedding at 6 dpi (Figure 2b). Cloacal shedding of pheasants did not increase daily as was seen with chickens, with vRNA first detected only at 2 dpi (n = 9/15), with peaks observed at 4 dpi and 6 dpi (Figure 2b), where 100% of remaining birds were shedding at 6 dpi (n = 4/4). It should be noted that a single bird had a high REU value at 4 dpi (#221, 2.13 × 10^6^ REU) which may account for these dual peaks.

The analysis of vRNA shedding from partridges initially demonstrated a sharp increase in the detection of vRNA, with buccal shedding detected in 20% at 2 dpi (n = 3/15) (Figure 2c), increasing until 4 dpi by which time 100% of birds were shedding (n = 13/13). From 4 dpi onward, the levels of detected vRNA decreased steadily until 10 dpi, where the last buccal shedding above the positive threshold was observed. Cloacal shedding in partridges followed a similar pattern to buccal shedding, with vRNA detected at 2 dpi (n = 1/15, 7%), with vRNA levels increasing until 4 dpi (n = 11/11, 100%), where again, levels of detected vRNA decreased until no vRNA was detected from the cloacal route by 10 dpi (Figure 2c).

To compare total shedding between species, area under the curve (AUC) analysis was carried out (Figure 2d,e). The analysis showed that there was a significant difference in shedding between chickens and pheasant via the cloacal route (*p* = 0.027), but no significant differences were observed elsewhere.

### 3.3. Serological Assessment of Birds Directly Infected with AV112-13

Only birds surviving to the end of the study were tested by HI, therefore serological analysis was only carried out on the two surviving partridges, both of which had seroconverted (Appendix A).

### 3.4. Tissue Distribution and Post-Mortem Examination of AV112-13 Infected Birds

The birds taken for PME had tissues analyzed for the presence of virus by the detection of viral antigen (by IHC). Fewer timepoints were available for chickens where birds were taken for planned PME due to the severity of clinical disease (1, 2, 3, 4 and 5 dpi), with only a single bird planned for PME remaining at 5 dpi.

Common gross pathology observations in chickens demonstrated that birds suffered from enlarged spleens and inflammation of the tissues of the upper respiratory tract, which persisted throughout the course of infection and was present at each post-mortem time point. As the clinical disease progressed, pinpoint hemorrhage of the proventriculus and some necrosis of the intestinal lymphoid mucosal tissue (MALT) were observed, accompanied by the typical green diarrhea associated with established NDV infection. Histopathological changes included the observation of macrophage sheath hyperplasia in the spleen along with rhinitis from 2 dpi, splenic and thymic necrosis, lymphocytic depletion in the bursa and spleen and tracheitis from 3 dpi. Encephalitis was observed at 4 dpi. Chickens with severe disease also showed necrosis in the cecal tonsils. 

Virus antigen immunohistochemistry on the tissues from PME chickens showed no labelling at 1 dpi (Table 1). Minimal labelling was noted in the spleen and turbinates of birds at 2 dpi. By 3 dpi, the birds retained positive labelling of virus antigen in the spleen and turbinates, but it was additionally observed in the feather follicle and trachea. At 4 dpi, the birds had antigen detected in the same tissues as 3 dpi, with the addition of the cecal tonsil and the brain. A bird humanely killed at 4 dpi and a planned PME at 5 dpi generally had increased distribution of staining in the same organs compared with the birds at 4 dpi, as well as in the cecal tonsil. Antigen detection was also observed in the bursa, which was not seen in the birds taken for PME at 1 to 4 dpi. Antigen detection examples on chicken tissues (4 dpi euthanized bird) are shown in Figure 3.

Due to the unknown severity of AV112-13 in pheasants and partridges, the birds were taken for planned PME at later staggered timepoints than the chickens. The rapid progress in disease severity following infection in pheasants meant that planned PMEs could only be undertaken at 2 and 4 dpi. However, pheasants were also examined by PME where clinical endpoints were met. Common post-mortem observations for the pheasants again included an enlarged spleen (from 2 dpi), with splenic necrosis being observed in samples from pheasants taken at 4 dpi. Lymphocytic depletion was also observed in the spleen and bursa, with necrosis of the MALT throughout the digestive tract from 4 dpi. Necrosis of the thymus, pancreas and liver, rhinitis and encephalitis were observed in 4 dpi and 7 dpi tissues with the severity of lesions being greater at the latter time point. Pancreatic necrosis and hemorrhages in the digestive tract were also observed at 7 dpi. Most animals were found to be carrying intestinal protozoal infections despite treatment with the recommended anti-protozoal drugs (Enrofloxacin and Toltrazuril) prior to the commencement of the study.

Analysis of antigen distribution in pheasants was carried out by IHC, with antigen being present at 2 dpi in the spleen and cecal tonsil (Table 1). Pheasants taken for PME at 4 dpi had increased labelling in the spleen and cecal tonsil in comparison to the 2 dpi birds whilst labelling in the intestine and turbinates was also observed. The tissues from birds terminated at 4 dpi due to increased clinical signs were also positive for antigen with increased frequency in the lung, trachea and turbinates. There was however a decrease in antigen detected in the bursa and thymus in the 4 dpi humanely killed birds. Interestingly, labelling was observed in the brain of the pheasants at day 4, commonly associated with virulent forms of AOAV-1 (Table 1 and Figure 3). A similar pattern of labelling was observed in the day 6 pheasant which succumbed to infection, with greater antigen detection intensity observed, including pronounced antigen signal in the brain (Table 1). Finally, a pheasant terminated for PME at 7 dpi again had detectable antigen in the same tissues as that detected in the 4 dpi pheasants, but with viral antigen also detected in the brain (Table 1/Figure 3). This latter feature correlates with the observation of clinical sequalae consistent with the infection of the central nervous system (e.g., loss of balance and paralysis) which were observed with this bird.

No specific gross changes to organs were observed for the infected partridges selected for PME. Histological changes consistent with proliferation of macrophage sheaths in the spleen were observed from 2 dpi. Necrosis of MALT in intestine was observed at 2, 4 and 7 dpi in some birds. All birds examined after 7 dpi showed non suppurative encephalitis. A small number of animals also had intestinal protozoa, again despite prior treatment with anti-protozoa drugs. As in the other species, tissue dissemination in partridges again appeared to be systemic, however, the histopathological evaluation of tissues appeared to demonstrate a lower viral antigen load when compared to that seen in the infected pheasants. The detection of antigen specific labelling in tissue sets harvested from partridges was limited, with only the cecal tonsil containing detectable antigen at 2 dpi, and the spleen, cecal tonsil, turbinates and thymus at 4 dpi (Table 1). Subsequently at 9 dpi (both PME birds and those humanely killed see Figure 3) and at 11 dpi, the only tissue with viral antigen signal was the brain and trachea (9 dpi), whilst no antigen was observed in any other tissues.

### 3.5. Assessment of Virus Transmission from Infected Birds to Naive In-Contact Birds

To initially examine the transmission dynamics of AV112-13 in the different bird species, two uninfected conspecifics were included with each species groups, at 1 dpi, to assess potential transmission following infection. Weights, clinical outcomes and swabbing was undertaken as per the directly infected birds. 

For the chickens, the analysis of daily weight demonstrated that both in-contact chickens initially gained weight, until 4 days post contact (dpc), after which weight loss was observed (Figure 4a). Clinical observations were similar to those seen in the directly infected chickens, albeit at delayed time-points (5 dpc). Due to the nature of the disease both in-contact chickens were humanely killed, at 6 and 7 dpc following the onset of clinical disease (Figure 4b). 

For transmission within the pheasant group to naive in-contact birds, a gain in weight was observed until day 6, when a sudden weight loss was observed in both pheasants (Figure 4a). Clinical observations of the in-contact pheasants following the onset of disease correlated with that seen in the directly infected pheasants, again, with a delay in onset of clinical disease (initially observed at 6 dpc). Both in-contact pheasants were humanely killed at 7 and 8 dpc, respectively (Figure 4b). Post-mortem examination of one of the in-contact pheasants was undertaken, with antigen detected in many of the tissues, including the trachea, turbinates, the spleen and the brain (Table 1). 

In-contact partridges maintained consistent weight gain until one bird showed a decrease in weight 13 dpc following the inoculation of the directly infected partridges but did regain weight by the end of the study (Figure 4a) and this bird had seroconverted by the termination of the experiment (Appendix A). However, the second in-contact partridge showed no weight loss and had not seroconverted, suggesting that the infection had been cleared before seroconversion occurred. Both in-contact partridges showed no other clinical signs and survived until the end of the study (Figure 4b).

The analysis of vRNA shedding from in-contact chickens demonstrated both buccal and cloacal shedding (4–6 dpc bird 1 and 5–7 dpc bird 2), respectively (Figure 4c). In-contact pheasants also shed vRNA at equivalent levels to their directly infected counterparts from both the buccal and cloacal routes, with one bird shedding from 3–8 dpc, whilst the second shed vRNA from 5–9 dpc (Figure 4d) at which point both birds were humanely killed as they had reached humane clinical endpoints. For the partridge group, in-contact partridges had detectable vRNA from the buccal and cloacal routes at 3–6 dpc (Figure 4e), but approximately 1 log_10_ lower than the peak of directly infected birds (Figure 2c), with shedding having ceased by 7 dpc in in-contact bird #1. Viral RNA was also detected from a second partridge between 4–11 days post contact of directly infected birds from the buccal route, the same bird that had weight loss (Figure 4a) and seroconverted (Appendix A), with vRNA levels slowly increasing and then decreasing, suggesting clearance of the virus. Interestingly vRNA was only detected on day 6 from the cloacal route of in-contact partridge #1. 

### 3.6. Analysis of Intra- and Inter-Species Transmission of AV112-13

To assess the ability of pheasants or partridges to transmit both to and between species, six pheasants and partridges were directly inoculated with 1 × 10^6^ EID50 of AV112-13 intranasally and/or intraocularly (D0). The following day, six naïve conspecifics were introduced (R1), with periodic sampling carried out to determine shedding of viral material and hence potential transmission to R1. Upon detection of vRNA in swabs taken from the R1 birds, D0 were removed, and six further naïve birds were introduced (R2). Swabbing from the buccal route of R2 was periodically carried out to determine the presence of vRNA. Again, once vRNA was detected in the R2 group, the R1 group were removed, and six naïve chickens were introduced (R3).

The analysis of the transmission within pheasants demonstrated that at 4 dpi, all six of the D0 pheasants had been successfully infected and were shedding vRNA. Examination of transmission in the R1 birds demonstrated transmission had occurred to two pheasants at 4 dpc, and to all six pheasants by 5 dpc (Figure 5a). Onward transmission to the R2 pheasants occurred by 4 dpc of R1 pheasants (n = 5) and to all six birds by 5 dpc (11 days since the D0 inoculation). The introduction of chickens (R3) continued the transmission with all six chickens shedding vRNA within 2 days of contact of the R2 pheasants (13 days since inoculation of D0 pheasants).

For the directly infected partridges (D0), all birds were shedding vRNA at 5 dpi (Figure 5b). Transmission of the virus occurred to the R1 partridges, with low level detection in three of the R1 partridges at 4 dpc. Indeed, the virus had been successfully transmitted to all six R1 partridges by 5 dpc. However, following the introduction of the R2 partridges, the virus was not transmitted to any of these introduced R2 birds. To confirm that virus transmission did not extend to the R2 birds, the R1 partridges were removed at 9 days post initial infection and six naïve chickens were introduced (R3). As with the R2 partridges, none of the R3 chickens had detectable levels of vRNA from the buccal route at 2 days post introduction, further demonstrating that no onward transmission occurred between R1 and R2 partridges.

## 4. Discussion

The ability for AOAV-1 viruses to infect and transmit between relevant species is a significant knowledge gap in our current understanding of these viruses. Here we have examined the pathogenesis and transmission of a highly virulent genotype VII.2 AOAV-1 (APMV-1/Chicken/Bulgaria/112/13) in chickens and game birds, to enable us to further understand the potential impact of this virus on game bird populations as well as the likely routes of incursion of AOAV-1 into the poultry sector. Although there have been numerous studies of the pathogenesis of genotype VII AOAV-1 in chickens, pathogenesis and transmission studies of these isolates in game birds are limited. We therefore wished to examine the role that game birds may play in the spread and transmission of a contemporaneously widespread virulent virus in the wild. The primary scope included both infection in different species with a defined virus dose and then later inter-species transmission. Finally, the ability to transmit from game birds to chickens was assessed. This approach was relevant as the semi-feral nature of game bird rearing means that there is potential for game birds to be a bridge between wild birds and commercial poultry.

The pathogenesis of AV112-13 was assessed in three avian species. The infection of both chickens and pheasants resulted in high morbidity and mortality of these two species. The spectrum of clinical disease was similar across these two species and supported previous studies assessing genotype VII.2 AMPV-1 viruses [18,31]. The analysis of the recent European genotype VII.2 virus initially isolated in Belgium in 2018 demonstrated 100% mortality seen in directly infected chickens after 7 days, as was seen within this study. Onward transmission was also observed in chickens with shedding seen at 3 dpi, with mortality observed in 5/6 birds in this study, and complemented transmission dynamics observed in this study with chickens (Figure 4a) and pheasants (Figure 4b and Figure 5a) [26]. Clinical disease was observed from 2 dpi in chickens and pheasants, with mean death times of 5.4 and 5.6 days, respectively. Shedding in these species was also evaluated through the detection of vRNA. Peak shedding in both chickens and pheasants coincided with the onset of clinical signs and mortality. Therefore, the asymptomatic shedding phase (day 1) has the potential for onward spread of AV112-13 within pheasant populations before disease is observed.

In contrast, the infection of partridges was associated with moderate clinical disease, overall reduced shedding of the virus (detection through viral nucleic acid) and limited pathology. Forty percent (n = 2/5) of directly infected partridges survived until the end of the study and both survivors seroconverted demonstrating productive infection. Where clinical disease occurred in partridges, it was delayed in onset by 3 days in comparison to chickens and pheasants, with mild presentation observed. The contrast in clinical picture and disease presentation between the chickens and pheasants and the partridges suggests that infection is better tolerated in this latter species. Interestingly, the partridges continued to show limited clinical disease even whilst vRNA was being detected in excreted material from the buccal and cloacal routes at 2 dpi, with peak shedding seen at 4 dpi, a day before initial clinical signs were observed. Similar observations have been made in studies assessing the infection of partridges with isolates of high pathogenicity avian influenza (unpublished data). Such differences in disease outcome may indicate that should an AOAV-1 be circulating in game birds, it is unlikely to be efficiently transmitted within some game bird species, however it may also circulate silently, and sub-optimal replication and shedding may prolong the risk of transmission and facilitate ongoing circulation within populations. The delayed and mild clinical presentation in partridges is of interest and requires further examination.

Alongside the assessment of pathogenesis across these species, transmission of this isolate in chickens, pheasants and partridges was undertaken through the addition of uninfected control birds to each group. In all three groups, both in-contact birds were successfully infected, demonstrating viral transmission, most likely through the shedding of infectious material from directly infected animals although exposure to infectious material within the local environment cannot be discounted. Indeed, the naïve in-contact birds not only became infected but also shed viral material from both buccal and cloacal routes. However, as with the directly infected birds, the clinical observations differed between the partridges, chickens and pheasants. Clinical signs in chickens and pheasants matched their directly infected counterparts. Mortality of these in-contact birds, as observed for chickens and pheasants that were directly infected, was 100%. In contrast, the in-contact partridges displayed limited clinical signs (one bird showed weight loss) and no mortality was observed. Shedding of vRNA was observed in both in-contact partridges, with one bird in particular shedding from the buccal route for a full seven days, the same bird where weight loss and seroconversion was observed. This observation may be of importance for infection in the field where low-level infection may occur. Only one of the in-contact partridges developed a neutralizing antibody response although innate-immune-based viral clearance post infection cannot be discounted. In assessing onward transmission within a species and between species, we have demonstrated that transmission from pheasant to pheasant occurs readily through two transmission stages and continued to in-contact chickens. In contrast, the transmission chain in partridges was only successful through one round with further transmission not being observed, even when chickens were introduced.

In conclusion, game birds are both susceptible to AOAV-1 genotype VII.2 infection and can shed infectious material and therefore represent a source of infection. The disparity in infection outcomes between the different species demonstrates differential susceptibility which may be restricted at the host level and requires further investigation. This data demonstrates that game birds may act as a bridge for AOAV-1 infection between wild birds and poultry due to their semi-feral rearing. This may represent a potential concern for commercial poultry holders wherever game birds may come into contact and would be influenced by population demographics including their density by species. Since the transport of game birds is common throughout Europe, there is a potential mechanism of AOAV-1 spread, as was reported in the 2005 UK outbreak in pheasants [14]. Due to the differing pathogenesis observed between the two game bird species, there is the potential that AOAV-1 could enter countries without being identified, until such time as they come into contact with a more susceptible species. Therefore, the testing of game birds, prior to these species crossing borders, for the presence of AOAV-1 may be beneficial in preventing disease incursions especially if infection pressure in a region with NDV is known to be high. However, this approach won’t mitigate against the threat of incursion from wild migratory birds and as such, an understanding of the infection dynamics of AOAV-1 across multiple species is warranted where these viruses pose a threat to poultry sector sustainability.

## Figures and Tables

**Figure 1 viruses-15-00536-f001:**
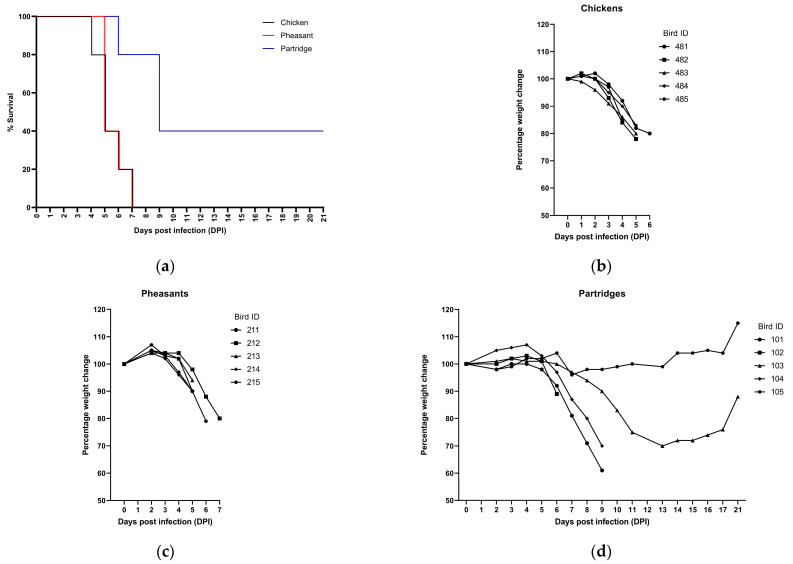
Weight changes and survival plots of chickens, pheasants and partridges infected with APMV-1/chicken/Bulgaria/112/13. (**a**) Survival kinetics of directly infected birds, chickens (black line), pheasants (red line) and partridges (blue line) following infection with APMV-1/chicken/Bulgaria/112/13. Weight changes as a percentage of their starting weights of (**b**) chickens, (**c**) pheasants and (**d**) partridges not selected for postmortem examination. Mortalities were determined for birds that were found dead or required humane killing on welfare grounds.

**Figure 2 viruses-15-00536-f002:**
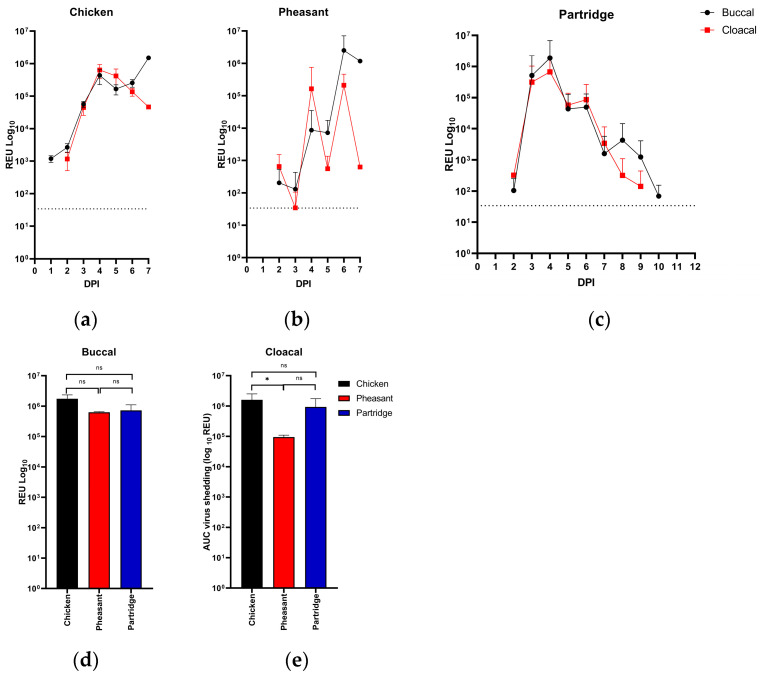
Mean shedding profiles of APMV-1/chicken/Bulgaria/112/13 in directly infected birds. Viral RNA detected from (**a**) chickens, (**b**) pheasants and (**c**) partridges of vRNA from buccal (black) and cloacal (red) swabs. Points show average vRNA detected as relative expressed unit (REU) of positive birds. Total shedding measured by area under the curve analysis of (d) buccal and (e) cloacal shedding. Error bars show standard deviation (SD), significant results (*p* < 0.05) designated by *, and dotted horizontal line shows limit of detection for (**a**)–(**c**).

**Figure 3 viruses-15-00536-f003:**
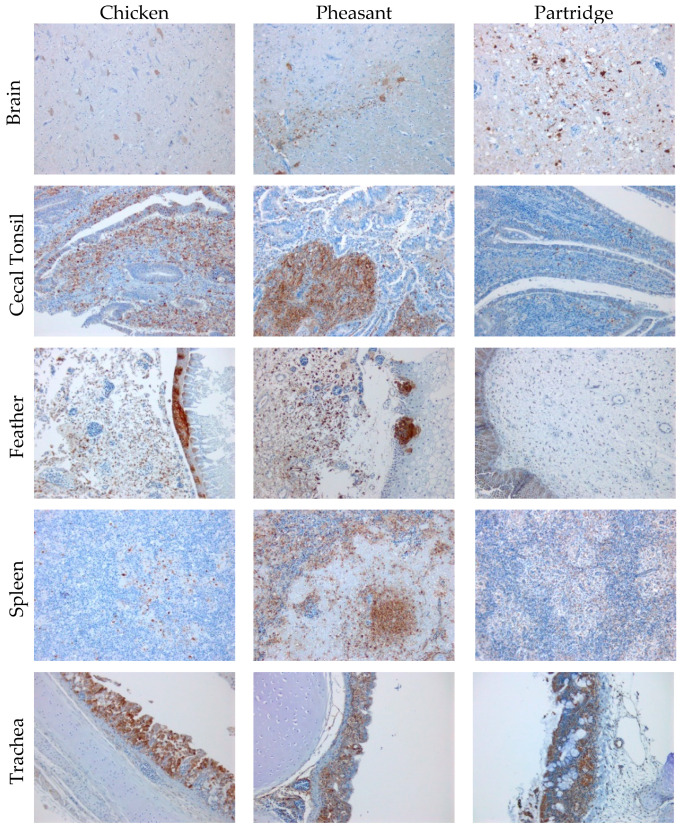
Representative immunohistochemical staining of chicken (4 dpi), pheasant (4 dpi) and partridge (9 dpi) tissues for AOAV-1 antigen following infection with APMV-1/chicken/Bulgaria/112/13.

**Figure 4 viruses-15-00536-f004:**
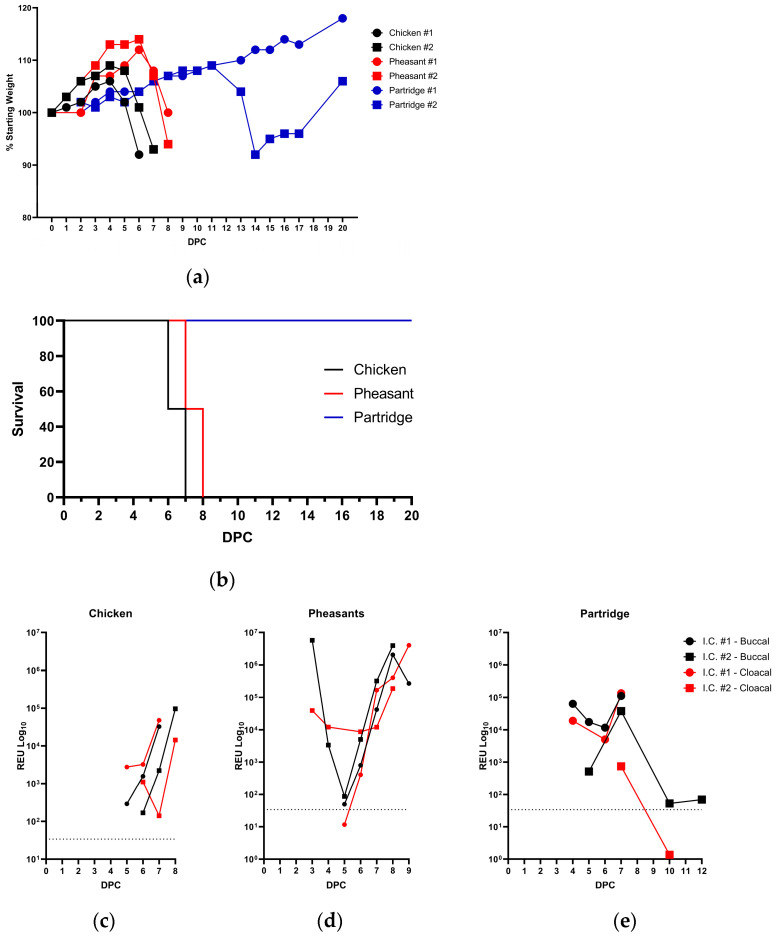
Weight changes, survival plots and viral RNA shedding of in-contact chickens, pheasants and partridges. Weight changes (**a**) as a percentage of their starting weights and survival kinetics (**b**) of in-contact chickens (black), pheasants (red) and partridges (blue). Mortalities were determined for birds that were found dead or required humane killing on welfare grounds. Buccal (black), or cloacal (red) viral RNA shedding of the in-contact chickens (**c**), pheasants (**d**) and partridges (**e**) expressed as relative expressed units (REU) and the dotted horizontal line showing limit of detection.

**Figure 5 viruses-15-00536-f005:**
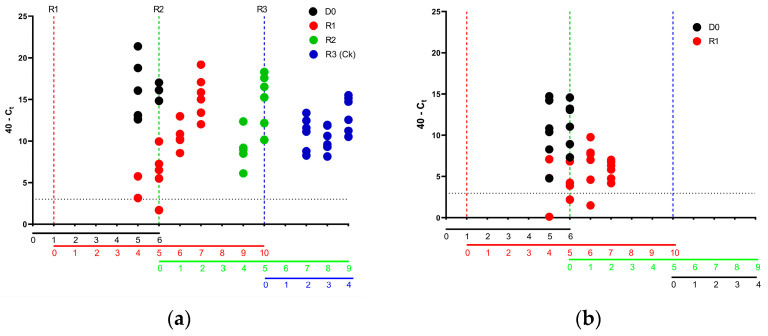
Intra-pheasant and Intra-partridge and onward transmission of APMV-1/chicken/Bulgaria/112/13. Viral RNA detected by buccal swabs and are shown as 40-Ct values for (**a**) D0 pheasants and (**b**) D0 partridges. The detected level for each individual bird is shown as D0 (black), R1 (red), R2 (green) and R3 (blue). The vertical dotted line shows the introduction of each transmission unit, whilst the horizontal line below shows the number of days’ contact for each group. The horizontal dotted line within the graph shows a positive cut-off for the L gene assay.

**Table 1 viruses-15-00536-t001:** Immunohistochemical distribution of AOAV-1 antigen (NP) in AV112-13 infected chickens, pheasants and partridges. NP staining is scored as +/- minimal, + few, ++ moderate, +++ numerous, NS—non-specific, ND—not determined. E—Euthanized; FD—Found dead, I.C. —In-contact conspecific.

	Chicken						Pheasant						Partridge					
	1 dpi	2 dpi	3 dpi	4 dpi	4 dpi (E)	5 dpi	2 dpi	4 dpi	4 dpi (E)	6 dpi (FD)	7 dpi (E)	9 dpi I.C.	2 dpi	4 dpi	7 dpi	9 dpi	9 dpi (E)	11 dpi
Heart	-	-	-	-	-	-	-	-	-	-	-	-	-	-	-	-	-	-
Feather	-	-	+	+	+	+	-	+/-	+/-	++	-	+/++	-	-	-	-	-	-
Skeletal Muscle	-	-	-	-	-	-	-	-	-	-	-	-	-	-	-	-	-	-
Spleen	-	+	+	+	++	ND	+	++	++	+++	+/-	++	+/-	++	-	-	+/-	-
Brain	-	-	-	+	+/-	ND	-	+/-	+/-	++	++	++	-	-	+/-	+	+	+
Kidneys	-	-	-	-	-	-	-	+/-	+/-	+	+/-	+	-	-	-	-	-	-
C. Tonsil	-	-	+/-	+	+++	++	+	++	++	++	+/-	++	+	+	-	-	+/-	-
Intestine	-	-	-	-	-	-	+/-	+	+	+	+/-	+	-	+/-	-	-	-	-
Lung	-	-	+/-	+/-	+/-		+/-	+/-	++	++	ND	++	-	-	-	-	-	-
Trachea	-	-	+	++	++	+++	NS	NS	++	+++	NS	++	-	-	-	-	+	-
Turbinates	-	++	++	++	++	++	NS	+	++	+++	NS	++	-	+	-	-	-	-
Liver	-	-	-	-	-	-	-	+/-	+	++	-	+/-	-	-	-	-	-	-
Bursa	-	-	-	-	+	++	+/-	++	+	+	+/-	++	-	+/-	-	-	-	-
Thymus	-	+/-	++	++	++	+++	+/-	+++	++	+++	ND	+++	-	+	-	-	ND	-
Pancreas	-	-	-	-	-	+/-	-	+/-	+	+	-	+/-	-	-	-	-	-	-

## Data Availability

Data is contained within the article or Appendix A.

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
