# Peer review of "Game Birds Can Act as Intermediaries of Virulent Genotype VII Avian Orthoavulavirus-1 between Wild Birds and Domestic Poultry"

_viruses, 2023, doi:10.3390/v15020536_

Round 1
Reviewer 1 Report
The authors studied pathogenesis and intra- and inter-species transmission of AOAV-1 genotype VII.2 in game birds (pheasants and partridges) and chicken. This study revealed that both pheasants and partridges may serve as intermediate hosts of AOAV-1 and may bridge the wild-birds - domestic poultry interface.
Main concerns
The main concerns relate to the study design. I wonder how the study was approved by an Animal Welfare and Ethical Review Body (AWERB)? It is noticeable that no negative controls were included, i.e. animals that were not infected but kept under the same conditions. No statistical calculations are given, neither for the group size calculation nor for the evaluation of the results.
The small-scale transmission study (L131 - 147) includes only two in-contact birds, which hardly allows for a statistical sound statement about the results, regarding the intra-species transmissibility. I wonder if this part of the study is relevant, in particular, if you monitor the same outcome in the second part of the study, where you used six in-contact birds with six directly inoculated birds.
Furthermore, there is a discrepancy between the number of animals reported in Materials and Methods (L131 - 133) (n = 15 per species) and the individuals shown in Figure 1 (n = 5 per species). With a total of 5 animals per species, I doubt that a valid conclusion is possible.
If the first experiment was more of a proof of concept, it is all the more surprising that in the second part (L148 - 157), the study design was not changed so that chickens were brought into direct contact with directly infected partridges. Instead, two further rounds of infection were carried out in partridges, although it was already clear that transmission within the partridge species was not very productive.
I wondered why you chose a genotype VII.2 originating from Bulgaria and not one of the genotypes responsible for the outbreaks in 2005 and 2006 (XIII.1.1 and VI respectively). But I noticed that a study with at least one of the two genotypes (VI.2) was conducted under the same license (70/8332) and has already been published (Craig S. Ross et al, (2023) Comparative pathogenesis of two genotype VI.2 avian paramyxovirus type-1 viruses (APMV-1) in pheasants, partridges and chickens, Avian Pathology, 52:1, 36-50, DOI: 10.1080/03079457.2022.2133680).
Minor concerns
Throughout the manuscript: Please use current taxonomic names according to ICTV (https://ictv.global/taxonomy/taxondetails?taxnode_id=202101591): Avian orthoavulavirus 1
L 179
No punctuation mark after the parentheses. Better a comma.
Figure 1
It would be more comprehensive to give the bird species in the figures b, c, d and not only in the figure legend.
L225 Correct the description: Weight changes … of a) should be b) chickens, …
Table 1
Describe I.C. in the table legend. Just for completion, not only describe +/- and +, ++, +++ but also –
L324
The statement “Interestingly, there was no labelling in the brain of either sets of day 4 pheasants” is not in accordance with table 1 and figure 3.
Figure 4 (c), (d), (e)
label the y-axis
L383/ 384
The suggestion, that infection had not occurred is somewhat misleading, when in the next paragraph you describe vRNA detection in buccal and cloacal swabs from both in-contact birds (L393).
Discussion
The data presented here should be discussed in the context of the already published study on the pathogenesis and transmissibility of genotype VI.2.
L522 – 524
Please delete.
Author Response
Rebuttal- Ross et al, 2022
We thank the reviewers for their constructive commentary to this manuscript. We have amended the manuscript in line with all comments. We believe that with these amendments the manuscript has been improved to a level worthy of publication in your esteemed journal. Please see comments against each point raised below, in bolded italics, whilst modifications within the manuscript are highlighted for easy of identification:
Reviewer 1:
The authors studied pathogenesis and intra- and inter-species transmission of AOAV-1 genotype VII.2 in game birds (pheasants and partridges) and chicken. This study revealed that both pheasants and partridges may serve as intermediate hosts of AOAV-1 and may bridge the wild-birds - domestic poultry interface.
Main concerns
The main concerns relate to the study design. I wonder how the study was approved by an Animal Welfare and Ethical Review Body (AWERB)? It is noticeable that no negative controls were included, i.e. animals that were not infected but kept under the same conditions. No statistical calculations are given, neither for the group size calculation nor for the evaluation of the results.
We thank the reviewer for this comment. Negative control birds are not required for this form of study in accordance with the 3Rs principles as birds are raised from specified pathogen free eggs in a sterile containment environment. The study was accepted with this study design after extensive internal review by our local ethics board.
The group sizes selected for experimentation were in line with previously published studies in the same area of research and as such consistency was required (Aldous et al., 2010 and Ross et al., 2023) Regarding results calculations, area under the curve analysis has been carried out for the 5 birds kept to the conclusion of the study to compare shedding data (now table 1) but no significant differences were observed between species.
The small-scale transmission study (L131 - 147) includes only two in-contact birds, which hardly allows for a statistical sound statement about the results, regarding the intra-species transmissibility. I wonder if this part of the study is relevant, in particular, if you monitor the same outcome in the second part of the study, where you used six in-contact birds with six directly inoculated birds.
As previously, we considered reduction (3Rs) when planning this experiment. The initial two in-contact birds were included to as a pilot for a further transmission study. We wanted to be sure that transmission did occur between game birds before the use of a further 48 birds (18 pheasants, 18 partridges and 12 chickens) which may have had a negative outcome regarding transmission.
Furthermore, there is a discrepancy between the number of animals reported in Materials and Methods (L131 - 133) (n = 15 per species) and the individuals shown in Figure 1 (n = 5 per species). With a total of 5 animals per species, I doubt that a valid conclusion is possible.
As birds were planned to be taken for PME throughout the study, the number of birds decreased throughout. We therefore only considered the 5 birds that were intended to be kept until the end of the study in this analysis as it would be unclear as to whether the birds taken for PME would have survived until the conclusion. We do state in L140 that the number of birds decreases over time
If the first experiment was more of a proof of concept, it is all the more surprising that in the second part (L148 - 157), the study design was not changed so that chickens were brought into direct contact with directly infected partridges. Instead, two further rounds of infection were carried out in partridges, although it was already clear that transmission within the partridge species was not very productive.
Although I agree that this may have been a more suitable experiment where we co-housed the chickens earlier with the partridges, we wanted to examine the transmission chain in partridges and whether there was subsequent onward transmission due to the negative result in the pilot study. Chickens were included after R2, as they appear to be much more sensitive to infection with genotype VII, so if there was excretion of infectious material from the partridges, then we would have hopefully observed clinical signs in the chickens.
I wondered why you chose a genotype VII.2 originating from Bulgaria and not one of the genotypes responsible for the outbreaks in 2005 and 2006 (XIII.1.1 and VI respectively). But I noticed that a study with at least one of the two genotypes (VI.2) was conducted under the same license (70/8332) and has already been published (Craig S. Ross et al, (2023) Comparative pathogenesis of two genotype VI.2 avian paramyxovirus type-1 viruses (APMV-1) in pheasants, partridges and chickens, Avian Pathology, 52:1, 36-50, DOI: 10.1080/03079457.2022.2133680).
This was a Defra (UK government) funded study where potential incursion into the UK would be assessed. You are correct that we have previously analysed two genotype VI.2 strains, we have also analysed the genotype XIII.1.1 from isolate from 2005 and this has also been written-up for a manuscript that is in preparation. This genotype VII.2 isolate was chosen as there was a number of outbreaks of this virus in the Mediterranean basin, and also recently in Belgium/Netherlands, and we therefore wished to examine the potential threat that this genotype may pose to the UK poultry system.
Minor concerns
Throughout the manuscript: Please use current taxonomic names according to ICTV (https://ictv.global/taxonomy/taxondetails?taxnode_id=202101591): Avian orthoavulavirus 1
Changed throughout the document. APMV-1 changed to AOAV-1 except when referring to the virus strain as this was how it is named in the laboratory.
L 179
No punctuation mark after the parentheses. Better a comma.
Modified
Figure 1
It would be more comprehensive to give the bird species in the figures b, c, d and not only in the figure legend.
L225 Correct the description: Weight changes … of a) should be b) chickens, …
Corrected. Also added “a) Survival kinetics….”
Table 1
Describe I.C. in the table legend. Just for completion, not only describe +/- and +, ++, +++ but also –
Added to the legend
L324
The statement “Interestingly, there was no labelling in the brain of either sets of day 4 pheasants” is not in accordance with table 1 and figure 3.
Apologies, this was an error on my part, there clearly is labelling in the pheasant brain on 4 dpi. The text (L 326) has been modified to reflect this
Figure 4 (c), (d), (e)
label the y-axis
Added to the figure
L383/ 384
The suggestion, that infection had not occurred is somewhat misleading, when in the next paragraph you describe vRNA detection in buccal and cloacal swabs from both in-contact birds (L393).
Modified this comment from “. However, the second in-contact partridge showed no weight loss and had not sero-converted, suggesting infection had not occurred” to “However, the second in-contact partridge showed no weight loss and had not sero-converted, suggesting infection had been cleared before sero-conversion occurred.”
Discussion
The data presented here should be discussed in the context of the already published study on the pathogenesis and transmissibility of genotype VI.2.
Added a section regarding recent genotype VII.2 outbreak in Belgium 2018.
L522 – 524
Removed, thank you for observing this
Reviewer 2 Report
I support this study. There is evidence of the NDV outbreaks in partridges or pheasants in nature. this is really important for the relevance of such a study. If we assume that hunters can carry the infection from them to places for domestic birds (chickens)
Please clarify the PCR diagnostics/primers for detecting of the infection.
Just remove “5. Conclusions 522 This section is not mandatory but can be added to the manuscript if the discussion is un- 523 usually long or complex.”
Author Response
Rebuttal- Ross et al, 2022
We thank the reviewers for their constructive commentary to this manuscript. We have amended the manuscript in line with all comments. We believe that with these amendments the manuscript has been improved to a level worthy of publication in your esteemed journal. Please see comments against each point raised below, in bolded italics, whilst modifications within the manuscript are highlighted for easy of identification:
Reviewer 2
I support this study. There is evidence of the NDV outbreaks in partridges or pheasants in nature. this is really important for the relevance of such a study. If we assume that hunters can carry the infection from them to places for domestic birds (chickens)
Please clarify the PCR diagnostics/primers for detecting of the infection.
The PCR primers and probe sequences are described in the reference (Sutton et al Development of an avian avulavirus 1 (AAvV-1) L-gene real-time RT-PCR assay using minor groove binding probes for application as a routine diagnostic tool. J Virol Methods 2019, 265, 9–14, doi:10.1016/j.jviromet.2018.12.001.)
Just remove “5. Conclusions This section is not mandatory but can be added to the manuscript if the discussion is un- usually long or complex.”
Removed – thank you for observing this.
Round 2
Reviewer 1 Report
All issues have been addressed sufficiently.